# *In Silico* Analyses, Experimental Verification and Application in DNA Vaccines of Ebolavirus GP-Derived pan-MHC-II-Restricted Epitopes

**DOI:** 10.3390/vaccines11101620

**Published:** 2023-10-20

**Authors:** Junqi Zhang, Baozeng Sun, Wenyang Shen, Zhenjie Wang, Yang Liu, Yubo Sun, Jiaxing Zhang, Ruibo Liu, Yongkai Wang, Tianyuan Bai, Zilu Ma, Cheng Luo, Xupeng Qiao, Xiyang Zhang, Shuya Yang, Yuanjie Sun, Dongbo Jiang, Kun Yang

**Affiliations:** 1Department of Immunology, Basic Medicine School, Air-Force Medical University (The Fourth Military Medical University), Xi’an 710032, China; zjq000211@163.com (J.Z.); sbz010115@163.com (B.S.); 18703383373@163.com (W.S.); wzj031129@163.com (Z.W.); sunyubo000103@163.com (Y.S.); jiaxingzhang09@163.com (J.Z.); 18391800515@163.com (R.L.); wangyongkaiyx@163.com (Y.W.); tianyuanbai@163.com (T.B.); 13930296799@163.com (Z.M.); lc1789702951@163.com (C.L.); qiaoxupeng2021@163.com (X.Q.); zhangxiyang199272@163.com (X.Z.); yangshuxiaoya@163.com (S.Y.); syjfly@163.com (Y.S.); 2Yingtan Detachment, Jiangxi Corps, Chinese People’s Armed Police Force, Yingtan 335000, China; 3Institute of AIDS Prevention and Control, Shaanxi Provincial Center for Disease Control and Prevention, Xi’an 710054, China; liuyangyang9610@163.com; 4The Key Laboratory of Bio-Hazard Damage and Prevention Medicine, Basic Medicine School, Air-Force Medical University (The Fourth Military Medical University), Xi’an 710032, China; 5Department of Rheumatology, Tangdu Hospital, Air-Force Medical University (The Fourth Military Medical University), Xi’an 710038, China

**Keywords:** Ebola virus glycoprotein (EBOV GP), pan-MHC-II epitope, immunoreactivity, in silico analysis, DNA vaccine, enzyme-linked immunospot (ELISpot) assay

## Abstract

(1) Background and Purpose: Ebola virus (EBOV) is the causative agent of Ebola virus disease (EVD), which causes extremely high mortality and widespread epidemics. The only glycoprotein (GP) on the surface of EBOV particles is the key to mediating viral invasion into host cells. DNA vaccines for EBOV are in development, but their effectiveness is unclear. The lack of immune characteristics resides in antigenic MHC class II reactivity. (2) Methods: We selected MHC-II molecules from four human leukocyte antigen II (HLA-II) superfamilies with 98% population coverage and eight mouse H2-I alleles. IEDB, NetMHCIIpan, SYFPEITHI, and Rankpep were used to screen MHC-II-restricted epitopes with high affinity for EBOV GP. Further immunogenicity and conservation analyses were performed using VaxiJen and BLASTp, respectively. EpiDock was used to simulate molecular docking. Cluster analysis and binding affinity analysis of EBOV GP epitopes and selected MHC-II molecules were performed using data from NetMHCIIpan. The selective GP epitopes were verified by the enzyme-linked immunospot (ELISpot) assay using splenocytes of BALB/c (H2d), C3H, and C57 mice after DNA vaccine pVAX-GP_EBO_ immunization. Subsequently, BALB/c mice were immunized with Protein-GP_EBO_, plasmid pVAX-GP_EBO_, and pVAX-LAMP/GP_EBO_, which encoded EBOV GP. The dominant epitopes of BALB/c (H-2-I-AdEd genotype) mice were verified by the enzyme-linked immunospot (ELISpot) assay. It is also used to evaluate and explore the advantages of pVAX-LAMP/GP_EBO_ and the reasons behind them. (3) Results: Thirty-one HLA-II-restricted and 68 H2-I-restricted selective epitopes were confirmed to have high affinity, immunogenicity, and conservation. Nineteen selective epitopes have cross-species reactivity with good performance in MHC-II molecular docking. The ELISpot results showed that pVAX-GP_EBO_ could induce a cellular immune response to the synthesized selective peptides. The better immunoprotection of the DNA vaccines pVAX-LAMP/GP_EBO_ coincides with the enhancement of the MHC class II response. (4) Conclusions: Promising MHC-II-restricted candidate epitopes of EBOV GP were identified in humans and mice, which is of great significance for the development and evaluation of Ebola vaccines.

## 1. Introduction

Ebola hemorrhagic fever manifests as multiorgan failure and shock, with a high mortality rate of 25–80% [1]. The pathogen Ebola virus (EBOV) contains five subtypes, including *Zaire*, *Bundibugyo*, *Sudan*, *Reston*, and *Tai Forest* [2]. As the first and most lethal of these subtypes, *Zaire* ebolavirus was discovered in 1976 and identified as the principal culprit in numerous outbreaks, even the notorious pandemic in West Africa in 2013–2016 [1,2,3].

EBOV is a negative-strand RNA virus belonging to the *Filoviridae* family. Its genome has seven genes encoding nucleoprotein (NP), viral protein (VP) 35, VP40, VP24, VP30, polymerase (L gene), and glycoprotein (GP) [4]. EBOV entry requires the surface glycoprotein (GP) to initiate attachment and fusion of viral and host membranes [5]. GP is widely believed to be the only viral protein on the surface of virus particles that mediates binding and invasion of host cells [6,7]. The interaction between EBOV GP and macrophages can induce robust innate and adaptive immune responses [8]. Most of the effective protective antibodies in the body fluids of Ebola virus disease (EVD) survivors are directed against EBOV GP [9]. Therefore, GP plays an important role in infection, elicits protective immunity, and is also considered an important target for vaccine research [10].

Long-term control of viral outbreaks requires the use of vaccines to confer acquired resistance and protection [11]. The development of the EBOV vaccine began in the 1970s, mainly in the form of a viral vector [12]. Only five EBOV vaccines have been approved, including Ervebo (FDA-approved), GamEvac-Combi (licensed in Russia), Zabdeno (approved in the EU), Mvabea (approved in the EU), and Ad5-EBOV (licensed in China) [13,14,15,16]. The development of conventional vaccines usually takes a long time. DNA and mRNA, as engineered antigen-coding products, could contribute to the rapid and effective development of vaccines against emerging pathogens worldwide today [12,17]. Previously, DNA vaccines encoding EBOV GP confirmed immune protection by targeting the MHC class II pathway [18].

It is universally acknowledged that the MHC class II presentation is required for the activation of Th cells and antiviral immunoprotection [19], yet it has rarely been reported on EBOV GP. Notably, a previous study confirmed the MHC class I presentation of EBOV GP by in silico analyses, exemplifying antigen immunoreactivity research [20]. In this study, the latest bioinformatics and traditional immunological methods were integrated to explore the pan-MHC-II-restricted immune reactivity of EBOV GP, and a series of selective epitopes were screened to aid in vaccine development.

## 2. Methods

### 2.1. Acquisition of Antigen Sequences

The sequence of EBOV GP (accession number: AY354458.1) was downloaded from NCBI GenBank, which is the target of various bioinformatics tools, including affinity prediction, conservation analysis, immunogenicity, molecular docking, and experiments.

### 2.2. Epitope Prediction

To obtain candidate epitope peptides unbiasedly, we integrated a variety of prediction tools to perform sequential molar oligopeptide segmentation of EBOV GP and affinity calculations between each peptide and MHC class II molecules. The MHC class II subtypes we selected are H2-Ab, H2-Ad, H2-Ak, H2-Aq, H2-As, H2-Au, H2-Ed, and H2-Ek of the H2-I genotype of mice. The human leukocyte antigen II (HLA-II) genotypes were selected from the four superfamilies HLA-DRB1, HLA-DRB3/4/5, HLA-DPA1/DPB1, and HLA-DQA1/DQB1, with a total population coverage of 98% (Appendix A). Peptide binding affinity to MHC class II molecules was predicted using four tools: IEDB-recommended (http://tools.iedb.org/mhcii/, accessed on 23 August 2021) [21], NetMHCIIpan (https://services.healthtech.dtu.dk/service.php?NetMHCIIpan-4.0, accessed on 23 August 2021; https://services.healthtech.dtu.dk/service.php?NetMHCIIpan-3.2, accessed on 25 August 2021) [22,23], SYFPEITHI (http://www.syfpeithi.de/bin/MHCServer.dll/EpitopePrediction.htm, accessed on 26 August 2021) [24], and Rankpep (http://imed.med.ucm.es/Tools/rankpep.html, accessed on 28 August 2021) [25]. Eventually, we chose the predicted epitopes that were ranked in the top 2% for IEDB recommended and NetMHCIIpan, highlighted in red for Rankpep, and in the top 2% for SYFPEITHI. The epitopes that were predicted by more than two prediction tools were subjected to sequential studies, and all the H2-I candidate epitopes with high affinity were selected for immunogenicity analysis.

### 2.3. Immunogenicity Analysis

Immunogenicity is related only to the peptide sequence and not to the spatial structure. We calculated the immunogenicity of the 15-peptide epitopes by VaxiJen 2.0 (http://www.ddg-pharmfac.net/vaxijen/VaxiJen/VaxiJen.html, accessed on 1 September 2021) [26]. The analysis results take a probability score >0.5 as the positive criterion, and the peptide is considered to be immunogenic; otherwise, it is not [27].

### 2.4. Conservation Analysis

To determine the degree of evolutionary conservation of the candidate antigenic epitopes among viral species sequences, we used the BLASTp (https://blast.ncbi.nlm.nih.gov/Blast.cgi, accessed on 13 September 2021) tool for conservation analysis of the predicted high-affinity 15-mer peptides. For the Ebola virus, the intraspecific conservative evaluation criteria are Zaire’s disease virus (taxid: 186538), excluding the 1995 Zaire virus (taxid: 128951). The evaluation and judgment of interspecies protection were confirmed to be Ebola virus (taxid: 186536), excluding Zaire disease virus (taxid: 186538). The epitopes conserved between humans (taxid: 9606) and mice (taxid: 10088) were excluded simultaneously. In the results of the analysis, if the E-value was <1 × 10^−5^, the sequence was considered conserved. The candidate epitopes can therefore be classified into four major categories according to conservation status: interspecies−intraspecies−, interspecies−intraspecies+, interspecies+intraspecies−, and interspecies+intraspecies+. “+” indicates that the epitopes were conserved, and “–” indicates that the epitopes were not conserved. Since there are no intraspecies nonconservative epitopes, “interspecies−intraspecies−” and “interspecies+intraspecies−” are not shown in the results table. Ultimately, we identified EBOV GP 15-mer peptides with high affinity for major HLA-II and H2-I subtypes that were immunogenic and evolutionarily conserved and thus named them pan-MHC-II selective epitopes.

### 2.5. Docking of pMHC Molecules

Molecular docking simulations were carried out to verify the combination of MHC class II molecules and selective epitopes *in silico*. Docking of MHC class II molecules and peptides was performed using EpiDOCK (http://ddg-pharmfac.net/epidock/EpiDockPage.html, accessed on 27 September 2021) [28], and the corresponding HLA-II isoforms were selected for docking simulation after inputting the sequences of 15 peptides. Each 15-peptide was subjected to docking simulation by seven 9-linked core amino acid sequences with the corresponding MHC class II molecules, so that the docking score was obtained. Fifteen peptides with a score higher than or equal to a given threshold were considered credible conjugates.

### 2.6. Epitopes and MHC-II Cluster

The polymorphism of MHC class II molecules and the diversity of epitope peptide amino acid sequences make the interaction between the two groups change thousands. Two-way hierarchical clustering by the algorithm TBtools was used to visualize the relationship between MHC-II subtypes and target antigen-related peptides [29]. Complete method-based two-way hierarchical clustering by Euclidean distance was used after the affinity ranking data were processed by base 2 logarithm and Z-Score minus. The higher the score, the stronger the affinity of the peptide to MHC-II molecules. MHC-II cluster analysis included 35 pan-MHC-II molecules interacting with 664 EBOV GP epitopes and was represented by a heat map.

### 2.7. Differences in Affinity between MHC II and EBOV GP Peptides

The affinity data between the peptide and MHC II are presented as Rank% of netMHCIIpan. When two MHC II subtypes are compared, each peptide with an affinity in the rank 2% of both is selected. The affinity of each genotype to the peptide was used for comparison. We took its logarithm base 2 as the data for analysis, and the lower the score was, the stronger the affinity between the peptide and the MHCII molecule. We also analyzed the case in individual mice from different strains. The chosen peptides had rank-2% affinity for all H2-I genotypes of both strains of mice. When comparing the affinity of strains with two H2-I genotypes, the score selected the one with the strongest affinity of the two genotypes for peptide. This actually underestimates its presentation power. Finally, 18 H2-Id-restricted high-affinity peptides were chosen to compare the affinity differences of each strain of mice.

### 2.8. Plasmid and His-Tagged Protein

Plasmids pVAX-LAMP, pVAX-GP_EBO_, and pVAX-LAMP/GP_EBO_ were all constructed in our laboratory [18], and no mutations were detected by Sanger sequencing. After sequencing, it was confirmed that there was no mutation. The plasmid was purified using the Plasmid Maxi kit (DP117, Chinese Tiangen) and stored at −20 °C until use. The construction, purification, and identification of His-tagged proteins were carried out according to the previous strategies of our research group [18].

### 2.9. Animals and Immunization

Female BALB/c, C57, and C3H mice at 6–8 weeks were purchased from the Laboratory Animal Centre of the Fourth Military Medical University. BALB/c mice were randomly divided into four groups, with 12 mice in each group, which were subcutaneously inoculated with PBS, pVAX-GP_EBO_ plasmid, pVAX-LAMP/GP_EBO_ plasmid, and Protein-GP_EBO_ at 0, 3, and 6 weeks. Our research group conducted extensive experiments in the early stages, which showed that the pVAX plasmid and pVAX-LAMP plasmid did not activate stronger immune responses than PBS [18,30,31,32]. Therefore, only PBS was used as a control in this study. C57 and C3H mice of 6 in each group were subcutaneously inoculated with pVAX-GP_EBO_ plasmid only, and 6 BALB/c mice were simultaneously immunized to observe the immune response in the three strains. The plasmid dose was 50 μg/mouse, and the Protein-GP_EBO_ dose was 10 μg/mouse. Two weeks after each immunization, tail venous blood was collected from the mice to separate the serum for the neutralization test. At 2.5 months after the last immunization, PBS, PVAX-GP_EBO_ plasmid, PVAX-LAMP/GP_EBO_ plasmid, and Protein-GP_EBO_ were injected into half of the mice in each group for an immune boost. The plasmid dose was 50 μg/mouse, and the protein dose was 10 μg/mouse. The mice were sacrificed 2 weeks after the immune boost to harvest spleen cells for the enzyme-linked immunospot (ELISpot) assay. Two mice were randomly sacrificed in each group for the experiment, and the experiment was repeated three times. The mice that received an immune boost were defined as post-boost in subsequent descriptions, while half of the mice that did not receive an immune boost were defined as pre-boost.

### 2.10. Peptides and ELISpot Assay

EBOV GP 15-mer peptides for artificial synthesis (ChinaPeptides, Shanghai, China) have high affinity for H-2-Id. Single peptides were diluted in PBS at 20 μg/mL for the ELISpot assay. Interferon-γ (IFN-γ) ELISpot reagents were obtained from BD Pharmingen Corporation (551083, BD Biosciences, New Jersey, USA) and used according to the manufacturer’s protocol.

Briefly, the ELISpot plates were coated with IFN-γ-specific capture antibodies and incubated overnight at 5 μg/mL (1:250) in sterile PBS at 4 °C. After the immunized mice were sacrificed, the spleen cells were washed and resuspended after erythrocyte lysis. RPMI-1640, containing 10% fetal bovine serum, was used for blocking at room temperature for 2 h. A total of 1 × 10^6^ spleen cells were added per well, with a final concentration of 20 μg/mL synthetic peptides. The negative control was pure RPMI-1640, and the positive control was Con A (10 μg/mL). The plates were cultured for 24 h in a 37 °C incubator with 5% CO_2_. After incubation, the plates were washed with ddH_2_O and PBST. Biotinylated rat anti-mouse IFN-γ antibody was added to each well and incubated at room temperature for 2 h. After washing with PBST, the cells were incubated for 1 h with 1:100 diluted streptomycin-HRP. After adding the substrate 3-amino-9-ethylcarbapene (AEC, DAKEWEI, Shenzhen, China), the reaction was stopped by washing with water. IFN-γ spots were counted after air-drying with a CTL ELISpot Reader (CTL, Kennesaw (Atlanta), GA, USA). Each experiment was performed in triplicate, and all results are averages of speckle formation cells (SFCs) minus that of the negative control in every 10^6^ spleen cells.

### 2.11. Serum Neutralization Test

The pseudovirus used for neutralization experiments was previously prepared by the research group and stored at −80 °C at a concentration of 10^6^-fold tissue culture infective dose (TCID50). The serum neutralizing antibody titer was detected by using a cell micro-culture neutralization test with monolayers of BHK-21 cells. BHK-21 cells were seeded into 96-well plates (3.5–4 × 10^4^ cells/well), cultured in DMEM with 10% fetal bovine serum, and incubated for 12 h at 37 °C. The mouse sera were diluted three times from the 1:10 initial dilution and then mixed with an equal volume of the 100-fold TCID50 rVSV-GPZEBOV pseudovirus. Each serum dilution was repeated in four wells. After incubation at 37 °C for 1 h, the mixture was added to the BHK-21 cell monolayer in a 96-well plate and incubated for 2 h at 37 °C and 5% CO_2_. The culture medium was discarded and replaced with a fresh medium. After incubation for another 24 h, virus infection was observed in each of the serum dilutions under a fluorescence microscope, and the 50% protective dose (PD50) was calculated by Karber’s method.

### 2.12. Data Visualization and Statistical Analysis

The heat map data for affinity and clustering are the Z-Score minus after-affinity ranking data, taking the logarithm base 2. TBtools was used to visualize the relationship between MHC-II subtypes and target antigen-related peptides [29], and complete method-based two-way hierarchical clustering was used for clustering analysis. We generated a bubble plot of the docking score using R 4.1.3. The value we use is the docking score minus the corresponding threshold. The color of the bubble indicates the optimal binding score of the epitope and MHC-II molecule, and the bubble size represents the number of cores that are able to dock with corresponding MHC-II molecules. GraphPad Prism 9.0.0.0 (121) was used to visualize the ELOSPOT results and the p-MHC affinity analysis. The mean and standard deviation of the three complexes in ELISpot were used to demonstrate the immune response of mice to each selective peptide. The Mann–Whitney test, one-way ANOVA, and paired *t*-test were used to evaluate the significance of differences between groups. The significance was marked with asterisks on the layout. A *p*-value < 0.05 was considered statistically significant.

## 3. Results

### 3.1. Screening of EBOV GP Peptides with High Affinity for Mouse H2-I and Major HLA-II Supertypes

After counting and excluding duplicate peptides using the process described in the Methods, 175 15-mer peptides of the H2-I subtype met the inclusion criteria (Table 1). H2-Ad-restricted 83 peptides of those were the most in H2-I genotypes, far more than the other eight H2-I genotypes. Eighty epitopes with high affinity in HLA-II subtypes were included (Table 1). HLA-DRB1 had the highest number of high-affinity peptides among the results for HLA-II subtypes.

### 3.2. Immunogenicity Analysis of EBOV GP 15-Mer Peptides

Peptides that can induce immune responses require not only high affinity but also immunogenicity. Thus, we performed immunogenicity analysis on the 15-mer peptides of EBOV GP. The results showed that 324 peptides, out of 664, were immunogenic. Forty-four of the 80 HLA-II subtype and 108 of the 217 H2-I subtype candidate epitopes with high affinity were immunogenic (Appendix A).

### 3.3. Conservation of EBOV GP MHC-II-Restricted Candidate Epitopes

Table 2 lists the results of the conservation analysis for all candidate epitopes with high affinity and strong immunogenicity. All high-affinity epitopes were intraspecifically conserved. For the interspecific conservation analysis, the number of conserved epitopes was greater than the number of nonconserved epitopes. A total of 31 15-mer peptides were selective epitopes for the human HLA-II subtype. The number of H2-I-restricted selective epitopes was 68. In addition, there were 19 selective epitopes that were cross-species reactive in humans and mice.

### 3.4. Interaction between Pan-MHC Class II Molecules and 9-Mer Epitopes

A heat map (Figure 1) was drawn to explore the binding feature between 15-mer peptides and the MHC-II subtype. This shows that the scale of binding affinity is regionally distributed. In general, there are six affinity hot spots in 34–38, 189–194, 209–219, 390–397, 512–516, and 612–630. The binding affinities of pMHC molecules are relatively weak in the epitopes residing at 76–83, 589–611, 632–641, and 658–664. Ordinarily, the binding patterns of the members of the same superfamily with the EBOV GP 15-mer peptide are similar. In particular, the presentation of EBOV GP peptides in HLA-DQA1* 0101/DQB1*0501 is closer to the HLA-DP superfamily. The binding affinity pattern of HLA-DPA1*0201/DPB1*1401 and the EBOV GP 15-mer peptide is distinct from that of other members of the HLA-DP superfamily. Two-way hierarchical clustering analysis was chosen for intuitive display.

Cluster analysis was performed on the affinity of 664 EBOV GP 15-mer peptides and 35 MHC-II subtype peptides (Figure 2). Thirty-five MHC-II genotypes are distributed into three clusters, including two cross-reactive clusters (HLA major and H2 major) and HLA-II-exclusive clusters. The similarity of EBOV GP presented by H2 and HLA-II reflects species cross-reactivity. H2-Ed, H2-Ek, HLA-DRB50101, and HLA-DRB11101 had similar results during antigen presentation. H2-Ab, H2-Ak, H2-Aq, H2-As, and H2-Au were similar to HLA-DQA/HLA-DQB in the presentation of EBOV GP. In the HLA-Ⅱ-exclusive cluster, compared with the HLA-DQ superfamily, the score of HLA-DQA1*0101/DQB1*0501 is more similar to that of HLA-DPA1/DPB1 (DPA1*0201/DPB1*0501, DPA1*0103/DPB1*0201, DPA1*0103/DPB1*0401, DPA1*0201/DPB1*0101, DPA1*0301/DPB1*0402). HLA-DPA1*0201/DPB1*1401 is assigned to cross-reactive clusters (HLA major), while other genotypes of HLA-DPA1/DPB1 are assigned to HLA-II exclusive clusters.

### 3.5. Selective Epitope Docking with Pan MHC-II Molecules

After the aforementioned analysis, we screened epitopes with high affinity, evolutionary conservation, and immunogenicity. It yielded 19 selective epitopes that were cross-species reactive. We further simulated molecular docking between peptides and MHC class II molecules. The results are shown in Figure 3. The red bubble in the figure indicates that there are 9-mer core sequences in the epitope that can stably assemble with MHC-II molecules for in silico verification. The larger the bubble, the more cores in the peptide that can closely bind to MHC-II molecules.

The docking scores of eight selective epitopes with all HLA-II subtypes have core docking bubbles with red color (scores > 0), which suggests that these selective epitopes (WVIILFQRTFSIPLG, VIILFQRTFSIPLGV, IILFQRTFSIPLGVI, DFAFHKEGAFFLYDR, GVVAFLILPQAKKDF, ELRTFSILNRKAIDF, LRTFSILNRKAIDFL, TEYLFEVDNLTYVQL) have core sites bound to all 23 selected HLA-II subtypes. These peptides can be considered to activate a wider range of virus-specific responses as potential protection for the population. In addition, five peptides (ILFQRTFSIPLGVIH, PPKVVNYEAGEWAEN, GLAWIPYFGPAAEGI, TTELRTFSILNRKAI, TELRTFSILNRKAID) had high docking scores with over 21 HLA-II subtypes. The protection activated by these five peptides is also widespread.

Figure 3 also shows that the HLA-II genotypes DPA1/DPB1 (DPA1*0103/DPB1*0201, DPA1*0101/DPB1*0501, DPA1*0102/DPB1*0602, DPA1*0301/DPB1*0302) and DRB1 (DRB1*0301, DRB1*0401, DRB1*0404, DRB1*0405, DRB1*0802, and DRB1*1501) generally have strong binding ability to all 19 selective epitopes, since they have at least one docking simulation above the threshold. All those genotypes may have stronger immune responses to EBOV.

### 3.6. Differences in Binding Affinity between MHC II and EBOV GP 15-Mer Peptides

Figure 4A shows the differences in the binding affinity between different H2-I molecules and 15-mer peptides from the same antigen. The results showed that all H2-I molecules exhibit differences in affinity for EBOV GP compared to others, with H-2-IAb and H-2-IAk showing the most significant differences compared to other H2-I molecules.

Individual mice have different affinities for the same antigenic peptides. We selected various H2 combinations of common inbred mouse genotypes to analyze their affinity for EBOV GP 15-mer peptides. Figure 4B shows the differences in binding affinity between diverse mouse H2-I genotypes and a series of 15-mer peptides of EBOV GP. The results showed that there were remarkable differences between H2-AkEk/Ab and others. Meanwhile, there was a significant difference in pMHC binding affinity between mice with H2-I as Ab (represented by C57) and AdEd (represented by BALB/c)/AkEk (represented by C3H). However, there was no significant difference in peptide affinity between BALB/c (H2d) and C3H (H2k) mice.

In addition, we compared the binding affinity between three mice with different MHC II genotypes and 18 synthetic 15-mer peptides, and the results are shown in Figure 4C,D. There was no significant difference in binding affinity between mice with the AdEd genotype and AkEk genotype, and the binding affinity of the two genotypes was better than that of Ab.

### 3.7. Experimental Verification

To verify the prediction, BALB/c, C57, and C3H mice were immunized with pVAX-GP_EBO_. Figure 5 shows that all three kinds of mice had immune responses to 18 15-mer epitopes with high binding affinity, but the results for BALB/c and C3H mice were more significant. The *p*-values of the two-way ANOVA for three groups and pairwise comparisons were less than 0.0001. Spots forming up to 130 were activated by candidate epitopes compared with the negative control. The results showed that BALB/c mice had strong immune responses to 18 15-mer epitopes, especially epitopes GYYSTTIRYQATGFG and GIRGFPRCRYVHKVS. The poor response in C57 mice could be attributed to the low affinity of its H2-Ib towards the H2-Id-restricted 18 epitopes, which is consistent with our findings in Section 3.6. This illustrates the accuracy of the prediction and verifies the immunoreactivity and immunogenicity of the relevant antigens.

### 3.8. Efficient Establishment of the Antiviral Immune Response

Previous studies have shown that the immune response induced by pVAX-LAMP is not significantly different from that induced by PBS [18]. The neutralizing antibody titers after three inoculations are shown in Figure 6. The novel packaged pseudovirus, rVSV-GPZEBOV, was determined to have an IU of 5 × 10^5^/mL. PBS-immunized mouse serum had no neutralizing activation. The serum neutralization titers of pVAX-GP_EBO_, pVAX-LAMP/GP_EBO_, and Protein-GP_EBO_ were significantly higher than those of the negative controls. The neutralizing antibody titers of the pVAX-LAMP/GP_EBO_ and Protein-GP_EBO_ groups were higher than those of the pVAX-GP_EBO_ group, and the difference was statistically significant. There was no significant difference in the serum neutralization titer between pVAX-LAMP/GP_EBO_ and Protein-GP_EBO_. Neutralization tests showed that the three vaccine groups had established immune responses, and pVAX-LAMP/GP_EBO_, and Protein-GP_EBO_ were stronger.

### 3.9. Evaluation of Cellular Immune Responses in Immunized Mice

To detect the activation of virus-specific T cells, the secretion of IFN-γ under the stimulation of 15 selected peptides at 3 months after the third immunization was detected through the ELISpot assay. The results showed that both pVAX-LAMP/GP_EBO_ and Protein-GP_EBO_ activated a stronger cellular immune response than PBS pre- or boost immune boost, demonstrating their long-term immune protection effect. The cellular immune response after the pVAX-GP_EBO_ immune boost is stronger than that of PBS. There was no significant increase in Protein-GP_EBO_, while the cellular response levels of pVAX-GP_EBO_ and pVAX-LAMP/GP_EBO_ increased with an immune boost. There were significant differences in IFN-γ secretion activated by peptides 9, 10, 11, 14, and 17 among different vaccines pre- or boost immune boost.

### 3.10. Immune Responses against 18 EBOV GP Peptides after Booster Immunization Were Significantly Improved

The spleen cells of the immunized mice were randomly sacrificed pre- and post-immune boost for the ELISpot assay. Figure 7 shows that the increased cellular immune response through immune boost was strongest in the pVAX-LAMP/GP_EBO_ group. That is, the enhanced immune response to these 18 H-2-Id-restricted peptides post-immune boost was more significant in the pVAX-LAMP/GP_EBO_-immunized group than in the pVAX-GP_EBO_ and Protein-GP_EBO_ groups.

### 3.11. BALB/c Mice Immune Responses to Corresponding H-2-Id-Restricted Epitopes Amplified with Immune Boost

From Figure 8A–C, the difference in the immune response level to each peptide in the pro-boost mice was smaller than that in the post-boost mice. As shown in Figure 8D–F, after the immune boost of pVAX-GP_EBO_ and pVAX-LAMP/GP_EBO_, differences in cellular immune responses of BALB/c to corresponding H-2-Id-restricted epitopes were amplified with the boost by the vaccines, and the difference in group pVAX-LAMP/GP_EBO_ was more significant than that in group pVAX-GP_EBO_ (Figure 8G).

## 4. Discussion

In this study, 31 HLA-II and 68 H2-I-restricted epitopes with high affinity and immunogenicity were screened by utilizing multiple algorithms, which were also interspecific and intraspecific conserved. Nineteen of them are cross-species reactive between humans and mice. Subsequent molecular docking preliminarily verified the feasibility and population reactivity of cross-reactive epitopes *in silico*. The characteristics of the population response and similarity between species were analyzed by bihierarchical affinity clustering of p-MHC-II molecules. Ultimately, high-affinity candidate epitopes effectively activated cellular immune responses in ELISpot experiments after immunization of BALB/c, C3H, and C57 mice. We screened and synthesized 18 EBOV GP 15-mer peptides by affinity to assess the extended MHC class II-restricted cellular immune response of EBOV GP vaccines. Neutralizing antibodies were used to evaluate the immune protection of candidate vaccines. Peptide-stimulated ELISpot assays and analyses determined the cellular immune responses at 3 months. Ultimately, pVAX-LAMP/GP_EBO_ showed the best immune protection and extended the MHC class II-restricted cellular immune response.

MHC class II molecules are critical in the control of many immune responses [33]. The affinity between MHC-II and peptide is one of the critical determinants of functional avidity [34]. It has already been demonstrated that higher-affinity peptides binding to MHC are necessary to activate T cell responses [35]. Previously, most epitope-related in silico studies relied on a single tool, which could lead to a bias in the analysis results. We adopted the “integration” strategy to obtain an accurate analysis of affinity between peptides and MHC-II, with the four algorithms complementing each other [36].

Peptides with high affinity do not always induce a cellular immune response. Immunogenicity also plays an important role in peptides inducing a strong immune response [20]. VaxiJen is the most widely used and highly cited server for immunogenicity prediction [26]. Between 2017 and 2021, a total of 275 studies used VaxiJen to analyze the immunogenicity of viral antigens. However, only 2.9% of them underwent subsequent experimental validation [37].

After comprehending the immunological properties, such as affinity, immunogenicity, and conservation, of each EBOV GP 15-mer peptide, the affinity characteristics between MHC class II molecules and the overall EBOV GP were considered. The results showed that H2-Ed, H2-Ek, HLA-DRB5*0101, and HLA-DRB1*1101 had similar scores when bound to peptides of EBOV GP. H2-Ab, H2-Ak, H2-Aq, H2-As, and H2-Au had similar results to HLA-DQA/HLA-DQB in antigen affinity. This phenomenon suggests that experiments carried out in those mice can partly reflect the response in humans, referring to the selection of experimental animals [38,39].

Molecular docking has been recognized as a valuable technique in computer-aided vaccine design [40,41]. The docking between 19 cross-species reactive epitopes and HLA-II molecules was simulated, suggesting that the affinity prediction results and the molecular docking calculation can be mutually validated. Integrated affinity analysis and molecular docking would make the results more accurate and convincing. Epitopes docking with multiple HLAs stimulate a broader immune response in vaccines [42,43], further indicating the feasibility of applying the 19 peptides to population immunity [44,45,46]. The potential of EBOV GP-derived pan-MHC-II epitopes in vaccine research and development was preliminarily explored.

ELISpot experiments after immunizing mice with a plasmid encoding GP were used to determine the availability of predictive epitopes. Owing to the limitations of human experiments, BALB/c mice were selected for the study. ELISpot experiments proved that the candidate epitope predicted based on the mouse H2-I subtype could indeed induce the immune response in mice. This finding hints at the immunoreactivity of predicted candidate epitopes restricted by human HLA-II subtypes. We further analyzed the affinity of individual mice for 18 peptides. BALB/c and C3H were similar and significantly higher than C57, which was preliminarily demonstrated by ELISpot. However, there are many restricted links between immune reactions activated by antigens, and bioinformatics technology cannot cover all of them. More “wet” experiments are needed to verify predictions in silico.

Previous studies have shown that LAMP can anchor the lysosomal membrane and be transported to the MHC II compartment (MIIC) [47,48]. Scholars have fused LAMP with the target antigen to build DNA vaccines and achieved protective effects [30,31,32,49], which have also been used in EBOV DNA vaccines [20]. However, its long-term MHC II-restricted immune response has not yet been elucidated.

Neutralizing antibody titers are one of the key indicators used to detect the antiviral efficacy of nucleic acid vaccines [50]. The MHC class II immune response can help activate B cells to produce neutralizing antibodies, which is effective for vaccines [51]. Thus, the neutralization antibody titer can reflect the presentation of MHC II laterally. DNA as a novel vaccine lacks clinical data. The subunit vaccine has been applied in the clinic for a long time and was added to the experimental system as a positive control. The DNA vaccines pVAX-GP_EBO_ without LAMP fusion were also compared simultaneously. Previous studies demonstrated that pVAX-LAMP/GP_EBO_ stimulated neutralization protection, which was well reproduced in our study. In the neutralization experiment, the antibody titer of pVAX-LAMP/GPEBO was significantly better than that of the pVAX- GP_EBO_ group, which was equivalent to that of Protein-GP_EBO_.

The immune protection of LAMP-fused DNA vaccines is equivalent to that of subunits and stronger than that of DNA vector carriers with only target genes. Protein vaccines, as exogenous antigens, also tend to be internalized, processed, and bound to class II molecules in lysosomal compartments [52]. Accordingly, the increased immunoresponse compared to pVAX-GP_EBO_ of pVAX-LAMP/GP_EBO_ and Proten-GP_EBO_ may be due to the targeting of MHC II molecules. Eighteen H-2-Id-restricted 15-mer peptides were selected, and then an ELISpot assay was used for verification. ELISpot experiments have made it capable of tracking antigen-specific T cells at the single-cell level [53]. The detection of IFN-γ reflects the reactivation of T helper type 1 (Th1) cells [54]. The cellular immune response of pVAX-GP_EBO_ post-immune boost is comparable to three instant immunizations [18], while pre-boost was similar to PBS. pVAX-LAMP/GP_EBO_ achieved a better cellular immune response 3 months later, both pre- and post-immune boost, and achieved the best response post-immune boost. Protein-GP_EBO_ performs the best pre-boost but shows little improvement post-boost. This suggests that pVAX-LAMP/GP_EBO_ has the strongest cellular immune response and may be applied in future defenses against the Ebola virus. The difference in response to peptides was amplified with the enhancement of immunity. This amplification is most significant in pVAX-LAMP/GP_EBO_. The better effect of candidate vaccines coincides with the enhancement of the MHC class II response.

Previous studies have shown that the improvement of the MHC-II response is of great significance to the effect of the EBOV DNA vaccine [18]. Based on the evaluation of EBOV GP pan-MHC-II-restricted immunoreactivity, the study analyzed and verified the selected epitopes on EBOV GP. The immunoreactivity of EBOV GP for H2-I genotypes, HLA-II subtypes, and populations was identified. This feature is essential for vaccine approaches, given the relevance of neutralizing antibodies against the pathogen and the support of helper T cells in prolonged antibody production. Through in silico immunoreactivity assessment, we can exploit peptides to design and evaluate vaccines. It is undoubtedly of great assistance in fighting against pathogenic microorganisms. In silico immunoreactivity assessment can be used to screen experimental animals to simulate human immunoresponse in the absence of genetically engineered mice.

## Figures and Tables

**Figure 1 vaccines-11-01620-f001:**
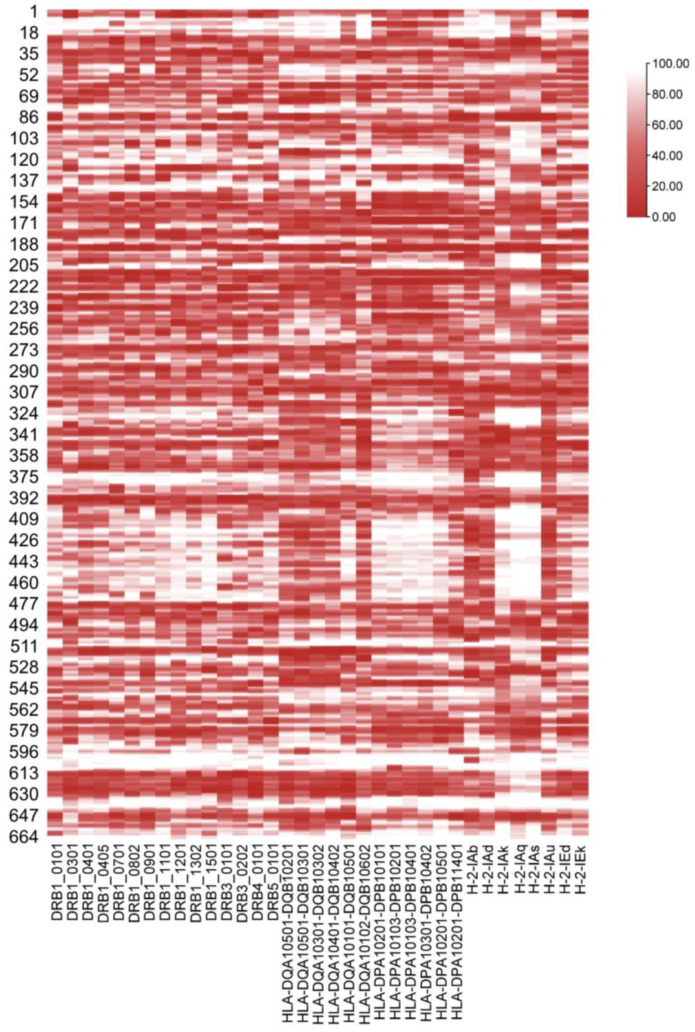
The binding affinity of different MHC-II subtypes and predicted 15-mer peptides. The ordinate is all 664 predicted 15-mer peptides, and the number is the position of the first amino acid of the 15-mer peptide in the GP sequence. The abscissa is the MHC-II subtype of mice and humans. The deeper the red color is, the stronger the affinity.

**Figure 2 vaccines-11-01620-f002:**
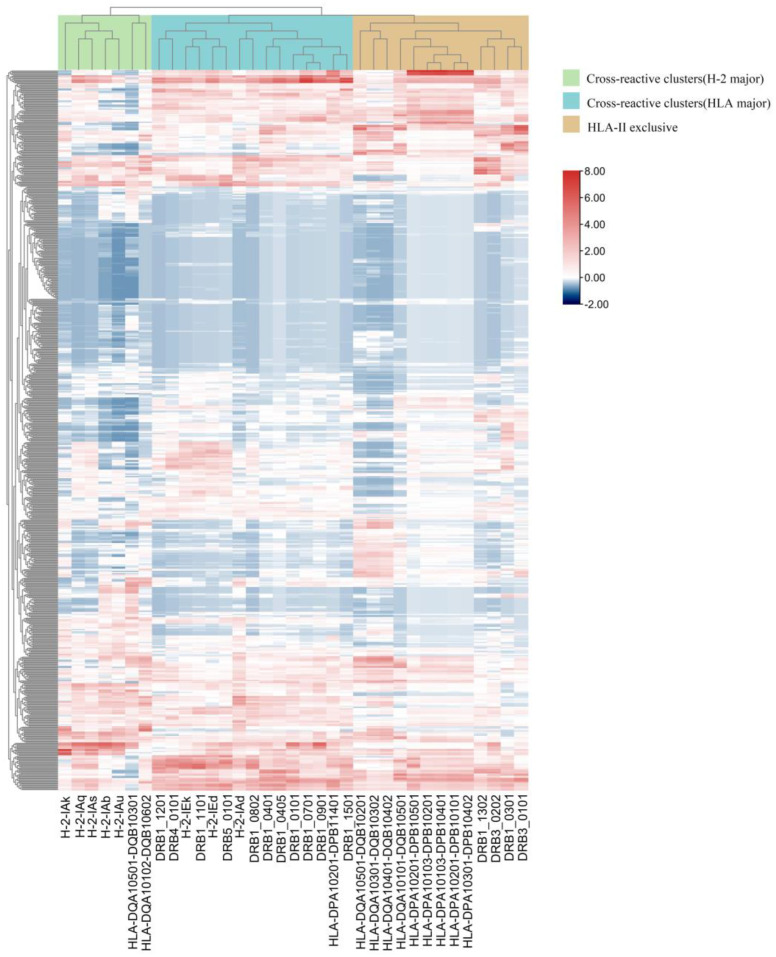
Hierarchical clustering of all EBOV GP 15-mer peptides binding to MHC class II. The abscissa is HLA-II and H2 genotypes, and the ordinate is peptides of EBOV GP. Red represents strong affinity, and blue represents weak affinity. The closer the distance, the more similar the pMHC interactions.

**Figure 3 vaccines-11-01620-f003:**
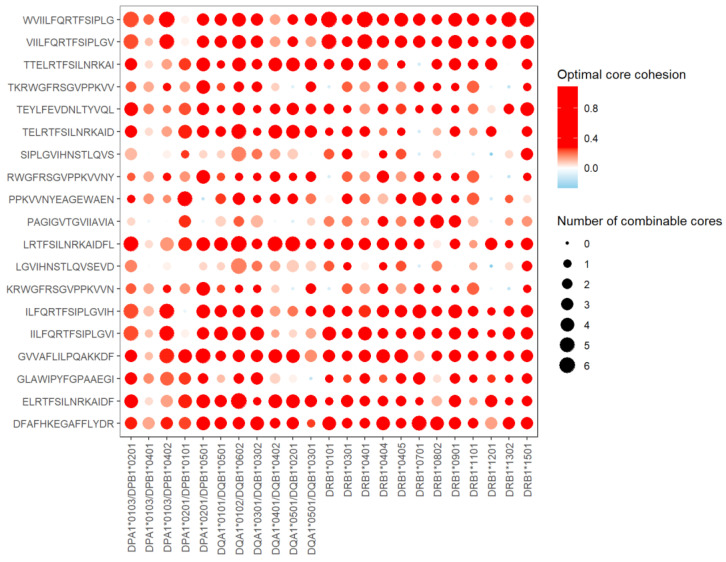
Docking score of the 9-mer peptide core of 15-mer peptides and HLA-II molecules. The docking results of 19 selective epitopes with MHC-II molecules were simulated. The 19 epitopes were all human and mouse MHC class II-restricted selective epitopes with high affinity, strong immunogenicity, and conservation (interspecific and intraspecific). Each MHC-II molecule docking with one peptide has seven docking scores of nine peptide cores. The darker the red color, the stronger the binding score. The blue color indicates that there is no 9-mer core docking with the HLA-II molecule. The larger the bubble, the more cores can truly bind with the HLA-II molecule.

**Figure 4 vaccines-11-01620-f004:**
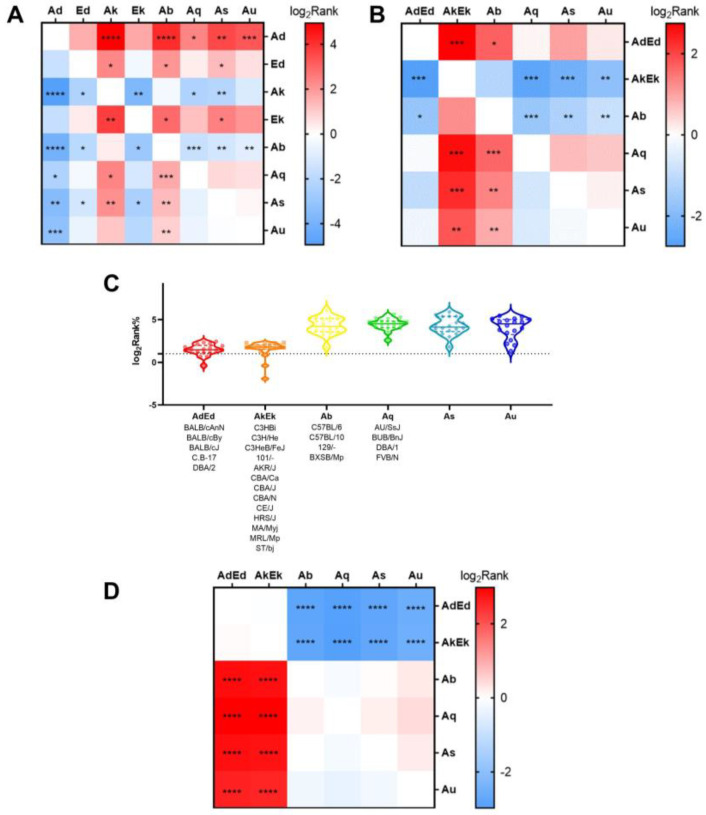
Differences in p-MHC binding affinity among H-2 genotypes and individual mice. In (**A**,**B**,**D**), red indicates that the mean p-MHC affinity of the H2 subtype at the top of the heat map is higher than the subtype on the corresponding edge, while blue indicates the opposite. The darker the color, the greater the gap. The H2-I is omitted for all MHC genotype names on the axes for aesthetics. (**A**) Comparison of the rank-2% peptides for the respective affinity of the genotypes. (**B**) Affinity comparison between H2-I genotypes of common strains of mice and the EBOV GP 15-mer peptide. (**C**) Affinity MHC genotypes for 18 peptides. The data points represent the binding affinities of H2-I molecules to specific peptides. The lower the value, the stronger the affinity. Its statistical analysis was visualized as (**D**). *, *p* < 0.05; **, *p* < 0.01; ***, *p* < 0.001; ****, *p* < 0.0001.

**Figure 5 vaccines-11-01620-f005:**
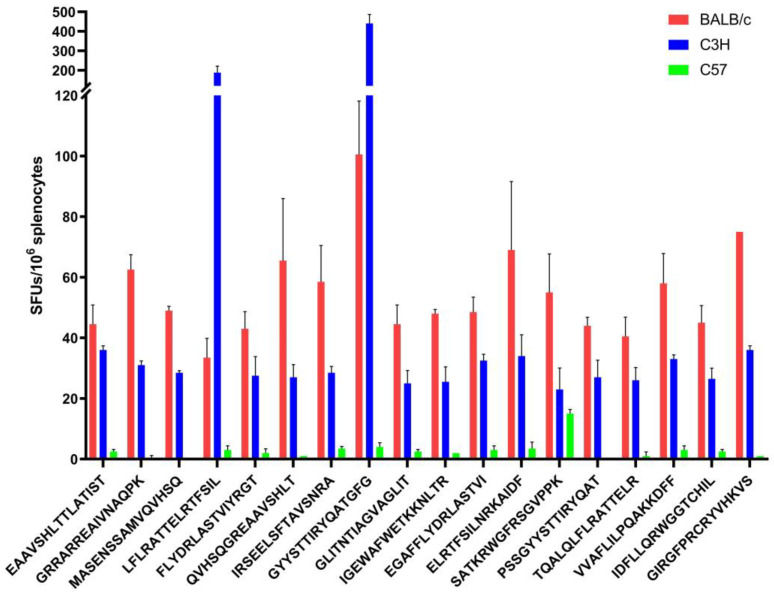
IFN-γ spot-forming units (SFUs) of BALB/c, C3H, and C57 mice to the 18 15-mer peptides of EBOV GP. ELISpot results of the three mice to the predicted 18 high affinity peptides of BALB/c mice after pVAX GP_EBO_ immunization.

**Figure 6 vaccines-11-01620-f006:**
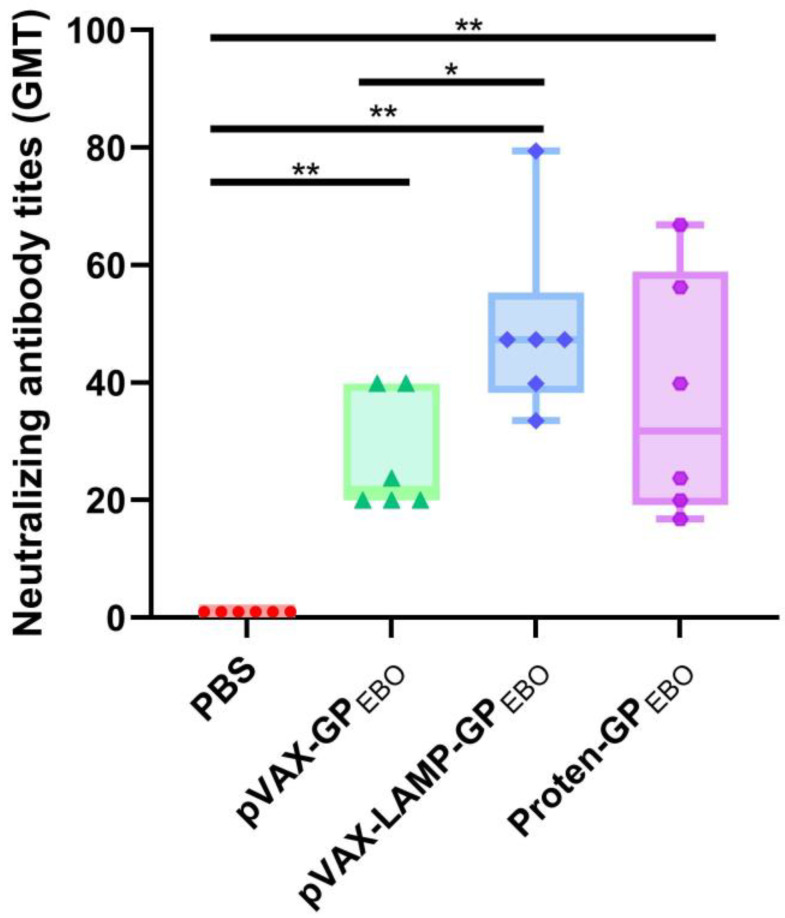
Neutralization titers in humoral responses were obtained by three consecutive immunizations of the three vaccines and the control. Symbols in the figure are the neutralizing antibody titers for each mouse in four groups. Neutralizing antibody titers were significantly higher in the three vaccine groups than in the control group. The neutralizing antibody titer of the pVAX-LAMP/GP_EBO_ group was higher than that of the other groups except Proten-GP_EBO_, and the differences were statistically significant. *, *p* < 0.05; **, *p* < 0.01.

**Figure 7 vaccines-11-01620-f007:**
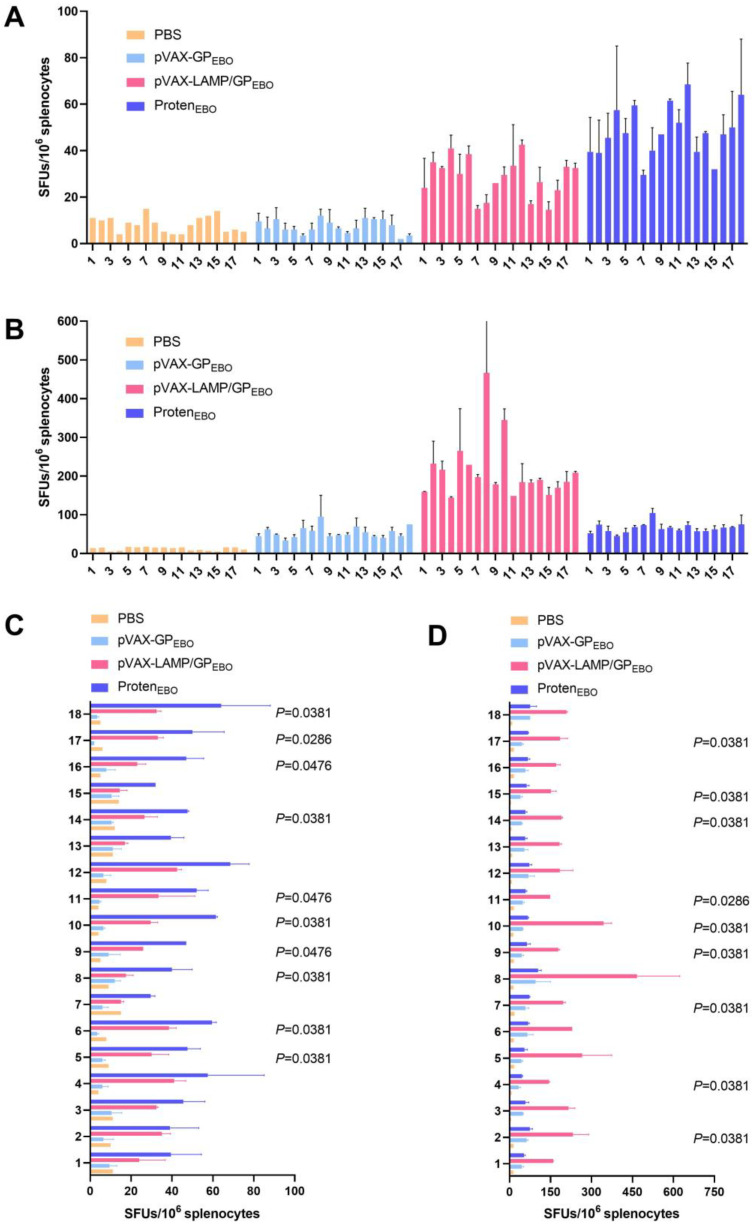
IFN-γ spot-forming units (SFUs) in BALB/c mice by the ELISpot assay. After BALB/c mice were immunized according to the prescribed procedure, antigen-specific spleen cytokine secretion was detected through the ELISpot assay. (**A**) IFN-γ secretion of mouse spleen cells detected by the ELISpot assay at 3 months after the third immunization. (**B**) IFN-γ secretion of mouse spleen cells detected by the ELISpot assay at 3 months after the third immunization. At 2.5 months after the last immunization, each group underwent a vaccine immunization boost once. Two weeks later, the secretion of IFN-γ was detected by the ELISpot assay. (**C**) Statistical analysis for (**A**). (**D**) Statistical analysis for (**B**).

**Figure 8 vaccines-11-01620-f008:**
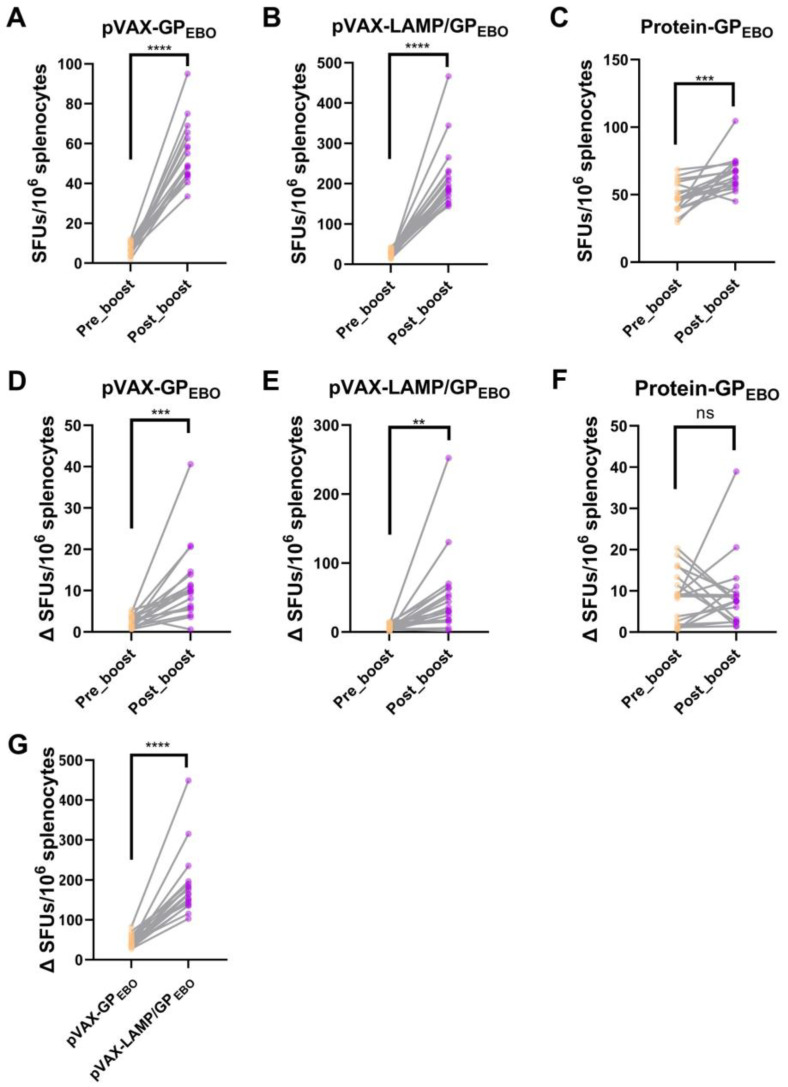
Comparison of the cellular immune response to all 18 15-mer peptides pre- and post-immune boost. (**A**–**C**) are *t*-test charts of ELISpot assay results of pVAX-GP_EBO_, pVAX-LAMP/GP_EBO_, and Protein-GP_EBO_ before and after booster immunization. The ordinate is the number of IFN-γ-secreting cells per 10^6^ vaccine-immunized mouse spleen cells after peptide stimulation, and the results of each peptide were compared correspondingly. The number of IFN-γ-secreting cells increased significantly after booster immunization with the three vaccines. The ordinates of (**D**–**F**) are the number of IFN-γ secreting cells minus the average value in every 10^6^ mouse spleen cells stimulated by each 15-mer epitope of 18 peptides, which indicates the difference in the immune response to each peptide in mice before and after booster immunization. (**G**) Amplification comparison between pVAX-GP_EBO_ and pVAX-LAMP/GP_EBO_ to 18 15-mer peptides pre- or post-immune boost. The response difference of each peptide pro-boost minus post-boost is made to obtain the value of the peptide. ns, *p* > 0.05; **, *p* < 0.01; ***, *p* < 0.001; ****, *p* < 0.0001.

**Table 1 vaccines-11-01620-t001:** Numbers of high-affinity MHC-II 15-mer peptides of EBOV GP.

MHC-Ⅱ Haplotypes	Prediction Tools	GP Epitopes	GP (Short Listed)
H2-A	IEDB	19 + 4(b, d)	121
	NetMHCⅡpan	5 + 1 + 5 + 2 + 3 + 3(b, d, k, q, s, u)	
	Rankpep	13 + 13 + 14 + 14 + 14 + 14(b, d, k, q, s, u)	
	SYFPEITHI	66 + 10(d, k)	
H2-E	IEDB	8(d)	89
	NetMHCⅡpan	4 + 3(d, k)	
	Rankpep	14 + 13 + 14 + 14(b, d, k, s)	
	SYFPEITHI	18 + 35(d, k)	
DRB1	IEDB	32	43
	NetMHCⅡpan	36	
	Rankpep	19	
	SYFPEITHI	26	
DRB3/4/5	IEDB	30	31
	NetMHCⅡpan	31	
	Rankpep	3	
	SYFPEITHI	0	
DQ	IEDB	15	19
	NetMHCⅡpan	15	
	Rankpep	8	
	SYFPEITHI	0	
DP	IEDB	13	8
	NetMHCⅡpan	13	
	Rankpep	0	
	SYFPEITHI	0	

The right column shows the number of peptides of mice MHC-II (H2-A, H2-E) and some alleles of major HLA-II supertypes (DRB1, DRB3/4/5, DQ, DPB1) predicted using indicated prediction tools based on EBOV GP sequences. Letters b, d, k, q, s and u are the alleles corresponding to mouse H2-I.

**Table 2 vaccines-11-01620-t002:** Conservation of the MHC-II-restricted candidate epitope of EBOV GP.

MHC-II Haplotypes	Interspecies− Intraspecies+	Interspecies+Intraspecies+
DRB1	13	30
DRB3/4/5	13	18
DQA1/DQB1	6	13
DPB1	0	8
H2-A	65	87
H2-E	19	63

## Data Availability

The animal study protocol was approved Experimental animal ethics committee of Air-Force Medical University (the Fourth Military Medical University). The submission process of the manuscripts meets all the review criteria. Readers interested in obtaining additional information beyond what is presented in the article can contact the first and correspondent authors.

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
