# Peer review of "In Silico Analyses, Experimental Verification and Application in DNA Vaccines of Ebolavirus GP-Derived pan-MHC-II-Restricted Epitopes"

_vaccines, 2023, doi:10.3390/vaccines11101620_

Round 1

Reviewer 1 Report

In the manuscript by Zhang and Sun et al., the authors show how immunity is achieved against EBOV glycoproteins in a vaccine regimen.  The authors begin with an in silico based approach then use the results to design experiments to test the predictions.  In general, the paper is well-written and support the authors’ conclusions.  However some sections and figures are unclear, which are my main points of criticism, detailed below.

Introduction – this section gives no background information on vaccination in general, let alone to EBOV.  There have been many reports on different vaccination regimens against EBOV, and these should be detailed in this section.

Figures 1-2 – the ordinate needs to be labeled.

Lines 285-287 – what is “serial analysis” and how were the 19 epitopes chosen?

Figure 4 – The explanation of the analyses performed here need to be greatly improved.  The methods section is sparse.  I went to the netMHCIIpan website, but could not replicate the analyses based on the methods presented. What are the abbreviations on the axes of panels A-B.  How are these selected using the algorithm.  What do the data points in panel C represent?  How are the peptides chosen in panels A-B and how are the data for specific peptides presented?

Line 347 – the authors state that there are immune responses for all three mice types, but there is little to no response for C57.

Figure 5 – what are “SUFs?”  Shouldn’t the y-axis be IFNgamma-producing cells/1e6 splenocytes?  What is the negative control in this experiment.  What statistical tests are performed?

Author Response

Dear Editor,

Thanks very much for taking your time to review this manuscript. We really appreciate all the comments raised by reviewers. Please find our itemized responses in below and re-submitted manuscript. Appended to this letter is our point-by-point response to the comments raised by the reviewers. The comments are reproduced and our responses are given directly afterward. We would also like to thank editor for allowing us to resubmit a revised copy of the manuscript.

Response to Reviewer 1

I would like to express my heartfelt gratitude for your recognition and approval of our work. Your feedback and suggestions have greatly improved the quality and clarity of the manuscript. We have taken your comments into careful consideration and have made the necessary revisions to improve the overall quality of our manuscript.

Comment 1: Introduction – this section gives no background information on vaccination in general, let alone to EBOV.  There have been many reports on different vaccination regimens against EBOV, and these should be detailed in this section.

Response: We appreciate the reviewer’s feedback regarding this issue. we have revised the introduction section to include a more comprehensive overview of vaccination regimens against EBOV.

Long-term control of viral outbreaks requires the use of vaccines to confer ac-quired resistance and protection.[PMID: 27686015] The development of the EBOV vaccine be-gan in the 1970s, which mainly in the form of viral vector.[PMID: 34329960] Only five EBOV vaccines have been approved, including Ervebo (FDA approved), GamEvac-Combi (li-censed in Russia), Zabdeno (approved in EU), Mvabea (approved in EU), and Ad5-EBOV (licensed in China).[PMID: 31591605; 33873076; 28152326; 27092831] The develop-ment of conventional vaccines usually takes a long time. DNA and mRNA as engi-neered antigen coding products, could contribute rapid and effective development of vaccines against emerging pathogens worldwide today.[PMID: 34329960; 33691133] Previ-ously, DNA vaccine encoding EBOV GP confirmed immune protection by targeting MHC class II pathway. [PMID: 34274417]

Comment 2: Figures 1-2 – the ordinate needs to be labeled.

Response: Thank you for your constructive comments. We labeled the ordinate of Figures 1 and supplement the other one as supporting material for Figure 2. In Figure 1, we have labeled the ordinate by indicating the intervals of 17 peptides. Meanwhile, in Figure 2, due to the application of bidirectional hierarchical clustering analysis, the peptides were redistributed and it became challenging to label the ordinate with intervals. Additionally, since the number of peptides is quite large, labeling all of them would result in an excessive length. In response to your suggestion, we would provide the specific peptide order here beneath our response and in the manuscript upon request. Readers who are interested in obtaining the clustered peptide order can contact the first and correspondent authors. We also included this information in the Data Availability Statement.

Comment3: Lines 285-287 – what is “serial analysis” and how were the 19 epitopes chosen?

Response: Thank you for pointing out the potential misunderstanding caused by the use of the term “serial analysis”. We have made the necessary changes in the revised version. The term “aforementioned” has been used instead to avoid any confusion.

Regarding the selection of the 19 epitopes, these epitopes were chosen based on aforementioned comprehensive analysis of their MHC affinity, immunogenicity, and conservation. In sections 3.1 to 3.3 of our results, we have described the screening process from affinity to immunogenicity to conservation. Through this rigorous analysis, we identified epitopes that possess high affinity and immunogenicity across species, ultimately resulting in the selection of both human and mouse specific epitopes, in other words, 19 cross-species selective epitopes.

Comment4: Figure 4 – The explanation of the analyses performed here need to be greatly improved.  The methods section is sparse.  I went to the netMHCIIpan website, but could not replicate the analyses based on the methods presented. What are the abbreviations on the axes of panels A-B.  How are these selected using the algorithm.  What do the data points in panel C represent?  How are the peptides chosen in panels A-B and how are the data for specific peptides presented?

Response: Thank you for your valuable feedback. We apologize for any confusion caused by the lack of detail in the methods section and the insufficient explanation of the analyses performed. We have addressed these issues by providing a more comprehensive description of the methods and a clearer explanation of the analyses in the revised manuscript. The legend on the axes of panels A is genotype of H2-I, which on panels B-D is H2 haplotype of inbreeding mice. Panel C shows the affinity MHC genotypes for 18 peptides. The data points represent the binding affinities of H2-I molecules to specific peptides. Regarding the peptides chosen in panels A-B, any two genotypes or haplotypes are compared, each peptide with an affinity in the rank 2% of involved H2-I genotype is selected. When an inbreeding mouse have more than one H2-I locus, the reactivity could be represented as the locus with stronger presentation ability. Thus, the topper rank between two loci for the peptide is selected to represent the inbreeding mice. The data example is as follows.

Comment5: Line 347 – the authors state that there are immune responses for all three mice types, but there is little to no response for C57.

Response: While we did observe immune responses in all three mouse types, the results for BALB/c and C3H mice were more significant. We agree with the reviewer’s observation and suppose that the poor response in C57 mice could be attributed to the low affinity of its H2-Ib towards the H2-Id restricted 18 epitopes, which is consistent with our analyses in section 3.6. Thank you for highlighting this aspect, and this explanation will be included in the revised manuscript.

Comment6: Figure 5 – what are “SUFs?”  Shouldn’t the y-axis be IFNgamma-producing cells/1e6 splenocytes?  What is the negative control in this experiment.  What statistical tests are performed?

Response: Thank you for pointing out the mistake in our manuscript. We apologize for the error in using “SUFs” instead of “SFUs”. We made the necessary correction and clarified the full spelling, IFN-γ spot forming units (SFUs), in the revised manuscript.

In this experiment, as described in section 2.10 of the methods, we included a negative control in each group where the stimulus was replaced with an equal volume of complete 1640 medium with 10% FBS. This control allowed us to assess the background levels of IFN-gamma production in the absence of the specific stimulus.

Regarding statistical tests, we performed two-way ANOVA for three groups and pairwise comparisons on this dataset. All p-values for the comparisons were found to be less than 0.0001. But we forgot to present the statistical tests in the manuscript, we will include them in the revised version. Thank you for bringing this to our attention.

We would like to express our gratefulness to the editors and referees. We have carefully reviewed and made every effort to address all the comments. We hope that the revised version meets the requirements of publication for the esteemed journal.

Best Regards,

Dr. Dongbo Jiang, superjames1991@foxmail.com

Prof. Kun Yang, yangkunkun@fmmu.edu.cn

Department of Immunology,

School of Basic Medicine, AFMU & FMMU,

169 Changle W Rd, Xi'an, China

86-18632105032

Reviewer 2 Report

The authors have done substantial work on the generation a peptide MHC based approach to arrive at peptide. It seems that they authors did rigorous studies to ascertain in-silico peptide MHC analysis and docking as well as proved within animals the response. While is is not clear whether there is any data on how the mice tolerated the immunization and as mentioned by the authors that all mice responded similarly, C57 clearly did not respond adequately. 

figure 8g I guess is a comparison between the GB and LAMP GB vectors, the research should change the representation of the graph to indicate bar charts only and not a connected line chart. if this is a pre boost vs post boost comparison then they should change the x axis representation. 

Authors should consider reworking the statistics of figure 8c since it doesn't look correct

fig 5,6,7, it is not clear ho many mice were taken, for the study, the authors should mention this

It is also not clear whether the peptides that were validated in figure 3 were used for further studies, it seems that the 18 15 mer peptides in figure 5 in the animal study. Please clarify this.

English looks fine but explanation of the story needs work

Author Response

Dear Editor,

Thanks very much for taking your time to review this manuscript. We really appreciate all the comments raised by reviewers. Please find our itemized responses in below and re-submitted manuscript. Appended to this letter is our point-by-point response to the comments raised by the reviewers. The comments are reproduced and our responses are given directly afterward. We would also like to thank editor for allowing us to resubmit a revised copy of the manuscript.

Response to Reviewer 2

I would like to express my heartfelt gratitude for your recognition and approval of my manuscript. Your feedback and suggestions have greatly improved the quality and clarity of my work. We have taken your comments into careful consideration and have made the necessary revisions to improve the overall quality of our manuscript.

Comment1: The authors have done substantial work on the generation a peptide MHC based approach to arrive at peptide. It seems that they authors did rigorous studies to ascertain in-silico peptide MHC analysis and docking as well as proved within animals the response. While is not clear whether there is any data on how the mice tolerated the immunization and as mentioned by the authors that all mice responded similarly, C57 clearly did not respond adequately.

Response: We apologize for any confusion caused in our manuscript. Thank you for pointing out the ambiguity. While we did observe immune responses in all three mouse strains, it is evident that the C57 mice exhibited a weaker response compared to the others. This is consistent with our previous research findings, where we also observed a significantly lower immune response in C57 mice compared to other strains at the same immunization dosage of Hantaan virus GP. We appreciate your suggestion and will consider investigating the differences in immune responses among inbred strains in our future studies. Thank you for bringing this to our attention.

Comment2: figure 8g I guess is a comparison between the GB and LAMP GB vectors, the research should change the representation of the graph to indicate bar charts only and not a connected line chart. if this is a pre boost vs post boost comparison then they should change the x axis representation.

Response: Thank you for your feedback and suggestions. We understand your concern about the clarity of the representation. As described in the figure caption, the connected line chart in Figure 8g represents the difference in response variability before and after boosting for each peptide in pVAX-GPEBO and pVAX-LAMP/GPEBO. The response difference of each peptide pro-boost minus post-boost is made to obtain the value of the peptide, then a paired T test was performed between the two groups. The purpose of showing the connected lines is to highlight the corresponding peptide pairs. We would like to maintain the current data presentation format as it effectively conveys the paired nature of the peptide comparisons. Your understanding would be sincerely appreciated.

Comment3: Authors should consider reworking the statistics of figure 8c since it doesn't look correct

Response: Thank you for pointing out the concern regarding the statistics of Figure 8c. We apologize for the mistake when marking the asterisk. We have carefully rework the data and ensure that the corrected statistics (p = 0.0005) are presented in the revised version of the manuscript.

Comment4: fig 5,6,7, it is not clear how many mice were taken, for the study, the authors should mention this

Response: Thank you for your valuable feedback. We appreciate the reviewer pointing out the lack of clarity regarding the number of mice used in the study in Figures 5, 6, and 7. In each ELISPOT assay, two mice were randomly selected from each group of 6 and sacrificed. The experiments were repeated three times to ensure accuracy and reproducibility. In a word, six mice in total were assigned in each group. We have now included this information in section 2.7 of the revised manuscript. Thank you once again for bringing this to our attention.

Comment5: It is also not clear whether the peptides that were validated in figure 3 were used for further studies, it seems that the 18 15-mer peptides in figure 5 in the animal study. Please clarify this.

Response: Thank you for raising this question. The peptides validated in Figure 3 is cross-species reactive between humans and mice. They were not used for further studies in the ELISPOT experiments with three reasons follow. Firstly, we focused on synthesizing high-affinity peptides specific to the BALB/c H2-Id genotype, which is commonly used for evaluating vaccines in animal studies. Secondly, we previously found that the immunoresponse of BALB/c is similar to humans’, and might reflect human immunoreactivity. Finally, we observed significant differences in the affinity of H2-Id restricted peptides among different mice to EBOV GP. It would benefit the validation to examine the responses of exactly one representative H2 genotype (such as H2-Id-restricted) peptides in various inbred mice. Indeed, 18 epitopes within ELISpot validations and vaccine evaluations got one identical and another 9 overlapping to the selected 19 cross-species epitopes in figure 3.

We would like to express our gratefulness to the editors and referees. We have carefully reviewed and made every effort to address all the comments. We hope that the revised version meets the requirements of publication for the esteemed journal.

Best Regards,

Dr. Dongbo Jiang, superjames1991@foxmail.com

Prof. Kun Yang, yangkunkun@fmmu.edu.cn

Department of Immunology,

School of Basic Medicine, AFMU & FMMU,

169 Changle W Rd, Xi'an, China

86-18632105032